# Effects of Ingesting Fucoidan Derived from *Cladosiphon okamuranus* Tokida on Human NK Cells: A Randomized, Double-Blind, Parallel-Group, Placebo-Controlled Pilot Study

**DOI:** 10.3390/md19060340

**Published:** 2021-06-15

**Authors:** Makoto Tomori, Takeaki Nagamine, Tomofumi Miyamoto, Masahiko Iha

**Affiliations:** 1South Product Co., Ltd., Okinawa 904-2311, Japan; miha@south-p.co.jp; 2Department of Natural Products Chemistry, Graduate School of Pharmaceutical Sciences, Kyushu University, Fukuoka 812-8582, Japan; miyamoto@phar.kyushu-u.ac.jp; 3Department of Nutrition, Takasaki University of Health and Welfare, Gunma 370-0036, Japan; Nagamine-t@kendai-clinic.jp

**Keywords:** fucoidan, *Cladosiphon okamuranus*, Okinawa mozuku, clinical study, NK cell activity, immunomodulatory

## Abstract

The aim of this study was to evaluate the effects of ingesting fucoidan derived from Okinawa mozuku (*Cladosiphon okamuranus*) on natural killer (NK) cell activity and to assess its safety in healthy adults via a randomized, double-blind, parallel-group, placebo-controlled pilot study. Subjects were randomly divided into two groups—a placebo group (ingesting citric acid, sucralose, and caramel beverages; *n* = 20; 45.5 ± 7.8 years (mean ± standard deviation)) and a fucoidan group (3.0 g/day from beverages; *n* = 20; 47.0 ± 7.6 years); after 12 weeks, blood, biochemical, and immunological tests were performed. Clinically adverse events were not observed in any of the tests during the study period. In addition, adverse events due to the test food were not observed. In the immunological tests, NK cell activity was significantly enhanced at 8 weeks in the fucoidan group, compared to before ingestion (0 weeks). In addition, a significantly enhanced NK cell activity was observed in male subjects at 8 weeks, compared with the placebo group. These results confirm that Okinawa mozuku-derived fucoidan enhances NK cell activity and suggest that it is a safe food material.

## 1. Introduction

Fucoidan is a general term for the sulfated polysaccharides contained in brown algae, the chemical structure of which differs depending on the seaweed species [1]. Edible seaweeds such as Kombu (*Kjellmaniella crassifolia*), Wakame (*Undaria pinnatifida*), and Okinawa mozuku (*Cladosiphon okamuranus*) are the main raw materials for fucoidan in Japan. In this study, fucoidan was extracted from Okinawa mozuku, which is an endemic species that grows only in the Okinawa Islands of Japan. Previous studies of Okinawa mozuku-derived fucoidan have reported its anticoagulant [2], anti-inflammatory [2], antiviral [3,4,5], anti-HTLV-1 [6,7], antitumor [8,9], antihepatitis [10], and antiulcer [11] effects. Okinawa mozuku-derived fucoidan has a chemical structure including a sulfate group and uronic acid bonded to the main fucose chain [12]. Although Okinawa mozuku-derived fucoidan has a high molecular weight, it is absorbed and has been detected in human blood and urine after ingestion [13,14,15]. In a previous study [16], we reported that the frequency of mozuku intake affects the absorption of fucoidan and the negativity of *Helicobacter pylori* antibody titers. Previous studies have also reported the immune effects of fucoidan. For example, we reported that Okinawa mozuku-derived fucoidan has an immune cell proliferative effect in mice, enhances macrophage phagocytosis, increases IgM, IgG, and IgA production, and suppresses IgE production [17]. Nagamine et al. [18] also reported that fucoidan activates natural killer (NK) cells in male cancer survivors.

In the present pilot study, we conducted a clinical trial using a placebo-controlled, randomized, double-blind, parallel-group comparison method. The primary outcome was to evaluate the effects of ingesting Okinawa mozuku-derived fucoidan on NK cells derived from healthy humans. The secondary outcome was to evaluate the safety of fucoidan consumption in healthy subjects under the observation of a medical doctor. In summary, adverse events due to the test food were not observed and Okinawa mozuku-derived fucoidan enhanced NK cell activity, especially in male subjects.

## 2. Results

### 2.1. Subject Background and Test Food Ingestion Rate

The background of the 39 subjects that completed the study is shown in Table 1. In the fucoidan group, the mean age (± standard deviation) of male and female subjects was 44.5 ± 8.0 and 46.5 ± 7.0 years, respectively; in the placebo group, these respective mean ages were 46.7 ± 6.5 years and 45.7 ± 7.1 years. The ingestion rate of the test food was 100.0%, 98.8%, 96.4%, 95.8%, and 86.9% in 33, 2, 2, 1, and 1 subject(s), respectively. The study started with 20 subjects in the fucoidan and placebo groups; however, one subject in the fucoidan group did not appear on the examination day after 8 weeks and could not be contacted. Therefore, in the fucoidan group, statistical analysis was performed with 20 subjects up to 4 weeks and 19 subjects after 8 weeks (Figure 1).

### 2.2. Immnologycal Assessment

#### 2.2.1. NK Cell Activity

In the fucoidan group, a significant increase in NK cell activity was observed at 8 weeks after ingestion when compared with NK cell activity at week 0 (Figure 2a). In male subjects in the fucoidan group, a significant increase in NK cell activity was observed at 8 weeks, relative to activity at week 0 in the placebo group, after fucoidan ingestion (Figure 2b). On the other hand, there was no significant effect of fucoidan on NK cell activity in female subjects (Figure 2c). In addition, NK cell activity results were described in Appendix A.

#### 2.2.2. Interferon-γ (IFN-γ) and Interleukin-2 (IL-2) Concentrations in the Blood

In tests of plasma IFN-γ concentrations, the number of fucoidan group subjects in which IFN-γ was detected (0, 4, and 8 weeks: 4 cases per week; 12 weeks: 6 cases) during the ingestion period tended to increase, whereas the number of placebo group subjects in which IFN-γ was detected (0 weeks: 5 cases; 4 weeks: 6 cases; 8 weeks: 3 cases; and 12 weeks: 4 cases) tended to decrease (Table 2). On the other hand, plasma IL-2 concentrations were below the detection limit in all subjects during the test period (data not shown). In addition, blood test results were graphed in Appendix A.

In addition, the number of detections in IFN-γ results were graphed in Appendix A. 

### 2.3. Safty Assessment

#### 2.3.1. Blood Tests

There were no significant differences in white blood cell count, hemoglobin, platelet count, or hematocrit between the placebo and fucoidan groups during the study period (Table 3). In addition, blood test results were graphed in Appendix A.

#### 2.3.2. Biochemical Tests

The results of the biochemical tests showed no abnormalities (Table 4). In the fucoidan and placebo group, compared with measurements at the same week, significant changes were observed at 12 and 16 weeks for magnesium (Mg), 8 weeks for iron (Fe). In addition, blood test results were graphed in Appendix A.

#### 2.3.3. Evaluation of Adverse Events

To evaluate the safety of the test food, adverse events during the ingestion period were recorded. Across both groups, 28 adverse events were observed during the ingestion period. However, following interviews and medical examinations by a doctor, it was concluded that serious adverse events had not occurred, and causal relationships with the test food did not exist. In addition, subjects recovered from all adverse events without further problems (Table 5).

## 3. Discussion

### 3.1. NK Cell Activity

The raw materials for fucoidan, i.e., edible seaweeds such as kombu, wakame, and mozuku, have a long history of being consumed as food in Japan. In the present study, fucoidan derived from mozuku (*C. okamuranus*) was used as a test sample. In previous studies, fucoidan has been reported to have effects on immunity in animals [17,19] and humans [18,20]. In the present study, healthy subjects were randomly allocated to fucoidan and placebo groups, and peripheral-blood-derived NK cell activity was evaluated after daily oral ingestion of fucoidan (3 g) for 12 weeks. Results showed that NK cell activity was significantly enhanced in the fucoidan group after ingestion of fucoidan; specifically, this activity was significantly enhanced in male subjects but not in female subjects. Normal cells have MHC class I receptors on their surface that suppress the activity of NK cells. However, cancer cells and virus-infected cells have decreased or deficient MHC class I expression, in which case the activation signal for NK cells is increased; this is a characteristic of attacking and removing cancer cells and virus-infected cells. In the present study, however, healthy adults were tested; thus, conditions related to cancer and virus-infected cells are not relevant. 

In general, the activation of NK cells requires the secretion of cytokines, such as IFN-γ, IL-2, and IL-12, by macrophages and helper T cells. In the present study, IFN-γ in the blood tended to increase over time in fucoidan group subjects, whereas IL-2 concentrations did not change significantly in any subject in either group during the ingestion and washout periods. Previously, Takahashi et al. [21] conducted an oral ingestion study of fucoidan in subjects with advanced cancer and reported that the production of IL-1β, IL-6, and TNF-α was significantly reduced 2 weeks after ingestion. In addition, Ohnogi et al. [22] reported that healthy subjects (one male and 14 female) who ingested fucoidan derived from Gagome kombu *(Kjellmaniella crassifolia)* for 4 weeks showed a significant suppression in the decrease of IFN-γ and IL-2 production. In a previous study in which fucoidan was administered to mice, NK cell activation significantly enhanced the production of IFN-γ [23]. Similarly, Murayama et al. [24] reported increased IFN-γ production in mice administered fucoidan, suggesting that the mechanism of action was via helper Th1 cell enhancement.

Previous studies have reported that macrophages are involved in the activation of NK cells by fucoidan. For example, we previously revealed that fucoidan significantly enhances IL-2 and IFN-γ levels in mice as well as macrophage phagocytosis [17]. Additional flows, other than those of cytokines, are involved in the activation of macrophages, e.g., radical scavenger receptors are involved. These receptors widely recognize negatively charged macromolecules such as low-density lipoproteins, lipopolysaccharides, and lipoteichoic acid [25,26]. Since fucoidan is a negatively charged polymer to which a sulfate group is bound, it is considered that fucoidan activity is enhanced via the radical scavenger receptor of macrophages. In addition, Miyazaki et al. [27] reported that fucoidan binds to the plasma membrane of macrophages to increase the production of nitric oxide and TNF-α. This suggests that the reaction is mediated by several pattern recognition receptors, such as Dectin-1, on the cell surface rather than through the phagocytosis of macrophages. Thus, it can be inferred that cytokines such as IFN-γ, which are produced by macrophages, are involved in the activation of NK cells by fucoidan. Overall, no significant change in IFN-γ concentration was observed in the present study; however, the cause of the seemingly fucoidan-mediated time-dependent increase in subjects with IFN-γ in the blood requires further investigation. 

Innate and adaptive immunity tends to be higher in women than in men, but the number of NK cells in men is higher than that in women [28]. Specifically, the normal range of NK cell activity is higher in adult males (post-puberty/adulthood) but becomes higher in females at old age [29]. In addition, hormones, genes, environment, age, etc. are also related to gender-specific differences in immune responses [30]. The mean age of the subjects in the fucoidan ingestion group in the present study was 47.0 ± 7.6 years, which is the period in which male NK cell activity is high. In addition, Nagamine et al. [18] conducted a fucoidan administration study in cancer survivors and reported that the NK cell activity of older male subjects (mean age: 73.9 ± 4.9; *n* = 11) was significantly higher than that of female subjects (mean age: 59.0 ± 7.7; *n* = 4). In a previous ex vivo study using ovariectomized rats [31], it was reported that fucoidan-treated NK cells had enhanced tumoricidal activity, but fucoidan-free standard diet-treated NK cells did not. This result suggests that fucoidan supplementation induces NK cell activity, regulating immunity caused by estrogen deficiency. In other words, female immune regulation is affected by organs such as the ovaries, and fucoidan may act on postmenopausal immunoregulation. According to these results, mozuku-derived fucoidan seems to enhance the activity of NK cells in males. The underlying mechanism of this effect, however, has yet to be evidenced and requires further investigation. In the future, we would like to increase the number of subjects and investigate the immunomodulatory effect of mozuku derived from fucoidan.

### 3.2. Safety Assessment

In the present study, fucoidan did not cause problematic adverse events when ingested at 3 g per day for 12 weeks. In addition, abnormalities were not detected in blood and biochemical tests. Although adverse events were occasionally observed during the study period, all were judged by the medical doctor to be unrelated to fucoidan ingestion, and patients recovered after the study. These results are consistent with mozuku, the raw material of fucoidan, being a type of seaweed that is often eaten in Japan and considered highly safe as a food item. Similar to the present results, Abe et al. [32] reported blood and urinalysis test results showing that ingestion of 4 g of mozuku-derived fucoidan daily for 2 weeks was safe. In addition, we previously reported no abnormalities in blood and biochemical tests performed on healthy Japanese adults following ingestion of 2 g of mozuku-derived fucoidan daily for 4 weeks [33]. High intake of fucoidan has been reported to cause diarrhea [7]; however, fucoidan intake in suitable amounts can improve defecation [34]. Diarrhea was not reported by the participants of the present study.

In conclusion, we have shown that Okinawa mozuku-derived fucoidan is safe as a food item and that it enhances NK cell activity, especially in males. In future research, we intend to investigate the effects of fucoidan on dendritic cells, macrophages, and cytokines (as examples) to elucidate the immunomodulatory action of these sulfated polysaccharides.

## 4. Materials and Methods

### 4.1. Materials

A beverage containing 1.5 g/50 mL of fucoidan derived from Okinawa mozuku (South Product, Uruma, Japan) was used as a sample for the fucoidan treatment group; citric acid and sucralose were added to the raw materials (which included 51.3% l-fucose, 18.8% sulfate ions, 14.4% uronic acid; mean molecular weight: 73.4 kDa). For the placebo group, citric acid and sucralose were blended in a beverage with caramel, which was used to ensure that the appearance of the placebo beverage did not differ from that of the fucoidan beverage.

### 4.2. Subjects

The study was conducted according to the Declaration of Helsinki. The study implementation plan, subject diary, and consent form were approved by the ethics review committee of Nihonbashi Cardiology Clinic, Tokyo, Japan (UMIN000043804), who also gave final approval to conduct the study. Moreover, the study was conducted under the guidance of a doctor at the Shinagawa Season Terrace Healthcare Clinic, Tokyo, Japan. The subjects were healthy men and women between the ages of 20 and 65 years old; each subject gave written consent after being provided with a sufficient explanation of the study. The target number of participants was set as the number required for statistical analysis. Subjects were recruited by KSO Corporation (Tokyo, Japan) and were assigned by block randomization to ensure that independent variables, e.g., age, gender, etc., did not differ significantly.

### 4.3. Study Design

The controller assigned subjects to two groups—the fucoidan and placebo groups—in a randomized, double-blind, parallel-group, placebo-controlled study (Figure 1). Both groups ingested two bottles containing samples per day. In a previous study [13], the amount of fucoidan detected in human blood and urine was 1 g/day. In addition, Abe et al. [32] administered fucoidan at a dose of 4.05 g/day, which was determined to be safe for human consumption; thus, the dose in this study was set to 3 g/day. The study was conducted from 17 July to 26 December 2016 (Figure 3).

### 4.4. Examination

All specimen measurements were performed by LSI Medience Corporation (Tokyo, Japan).

#### 4.4.1. NK Cell Activity

NK cell activity was measured in the K562 cell line (Dainippon Pharmaceutical, Japan) labeled with ^51^Cr using a cytotoxicity test. The NK cell activity test used subject blood on the day of first ingestion (week 0) and after 4, 8, 12, and 16 weeks. Blood samples were collected into heparinized tubes. After centrifugation of the blood samples with a lymphocyte separation medium, interface mononuclear cells were collected and suspended in RPMI-1640 medium supplemented with 10% fetal bovine serum (FBS). The samples were centrifuged again, and mononuclear cells were collected and mixed with ^51^Cr-labeled target cells (K562) at a ratio of 50:1. The cell mixture was cultured at 37 °C and 5% CO_2_ for 4 h. The ^51^Cr released from the target cells by NK cell cytotoxicity was determined using a gamma counter (ARC370, Hitachi Aloka Medical, Mitaka, Japan). The percentage of cytotoxicity was calculated as follows: cytotoxicity (%) = (experimental ^51^Cr release − spontaneous ^51^Cr release)/(maximal ^51^Cr release − spontaneous ^51^Cr release) × 100.

#### 4.4.2. INF-γ and IL-2 Concentrations in the Blood

IFN-γ (Human IFN-gamma Quantikine ELISA Kit, R&D Systems, Minneapolis, MN, USA) and IL-2 (Human IL-2 Quantikine ELISA Kit, R&D Systems, Minneapolis, MN, USA) concentrations in the plasma of subjects were measured using sandwich ELISA on the day of first ingestion (week 0) and after 4, 8, 12, and 16 weeks.

#### 4.4.3. Blood Tests, Biochemical Tests, and Safety Assessment

Blood tests (white blood cells, red blood cells, hemoglobin, hematocrit, and platelets) and biochemical tests (aspartate aminotransferase, alanine aminotransferase, LDH, T-bil, alkaline phosphatase, γ-glutamyl transpeptidase, creatine kinase, fasting blood sugar, HbA1c, total cholesterol, low-density lipoprotein cholesterol, high-density lipoprotein cholesterol, triglyceride, total protein, albumin, urea nitrogen, CRE, uric acid, sodium, chlorine, potassium, calcium, inorganic phosphorus, magnesium, and iron) were performed on the day of first ingestion (week 0) and 4, 8, 12, and 16 weeks later. In addition, during the same period, a doctor interviewed and examined the subjects as a safety evaluation.

### 4.5. Statistical Analysis

Data are presented as means ± standard deviation. Data were analyzed using Dunnett’s multiple-test within groups and Welch’s *t*-test between groups. *p* < 0.05 was considered to indicate statistical significance. Statcel version 4 software (OMS Publishing, Japan) was used to conduct all statistical analyses.

## Figures and Tables

**Figure 1 marinedrugs-19-00340-f001:**
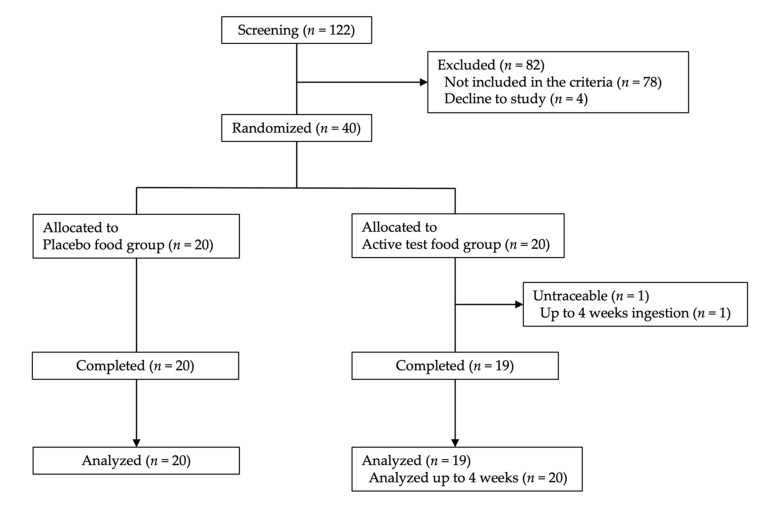
Flow diagram of the clinical study procedure. Values in parentheses show the number of participants (*n*).

**Figure 2 marinedrugs-19-00340-f002:**
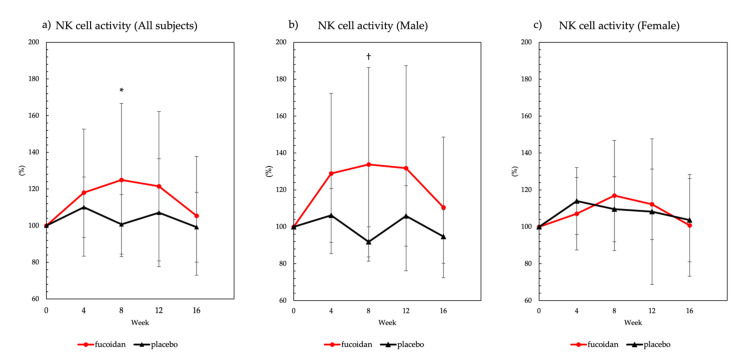
Natural killer (NK) cell activity. NK cell activity was expressed as relative to basal value (100%). All data were presented as mean ± standard deviation (*n* = 20 subjects per group). NK cell activity in (**a**) all subjects, (**b**) males, and (**c**) females. * Significant difference compared with week 0 within the fucoidan group (*p* < 0.05, Dunnett’s multiple test). ^†^ Significant difference between the placebo and fucoidan groups within the same week (*p* < 0.05, *t*-test).

**Figure 3 marinedrugs-19-00340-f003:**
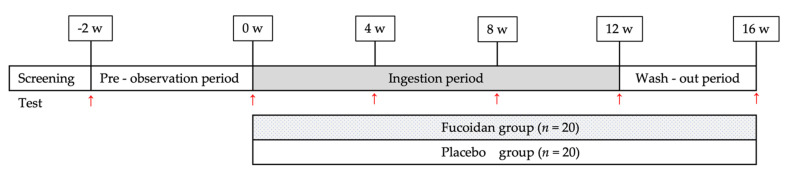
Overall study design. w: week; red arrows: testing of subjects; *n*: number of subjects.

**Table 1 marinedrugs-19-00340-t001:** Background of subjects in the placebo and fucoidan groups in the present study.

Parameter	Placebo	Fucoidan	*p*-Value
Subjects (*n*)	20	20	1.00
Male (*n*)	10	10	1.00
Female (*n*)	10	10	1.00
Age (years)	45.5 ± 7.8	47.0 ± 7.6	0.55
Body weight (kg)	59.1 ± 9.5	57.6 ± 9.8	0.63
Body mass index (kg/m^2^)	21.7 ± 2.3	21.0 ± 2.6	0.39

Age, body weight, and body mass index data are presented as mean ± standard deviation (*n* = 20 subjects per group). *n*: number of subjects; *p*: *p* value (placebo vs. fucoidan).

**Table 2 marinedrugs-19-00340-t002:** Number of detection in IFN-γ.

Group	0 w	4 w	8 w	12 w	16 w
Fucoidan	4	4	4	6	3
Placebo	5	6	3	4	3

Between-group comparison of subjects in which IFN-γ concentrations were detected in the blood. Detection limit: >1.56 pg/mL.

**Table 3 marinedrugs-19-00340-t003:** Blood test results.

Parameter	Group	0 w	4 w	8 w	12 w	16 w
WBCs (μL)	Fucoidan	5720 ± 1233	5080 ± 1307	5179± 1280	5168 ± 1433	5589 ± 1610
	Placebo	5950 ± 1674	5535 ± 911	5660 ± 985	5520 ± 1282	5685 ± 1132
RBCs (×10^4^/μL)	Fucoidan	453.6 ± 41.4	458.2 ± 37.5	453.3 ± 36.4	456.8 ± 40.2	463.6 ± 37.4
	Placebo	459.8 ± 28.3	462.7 ± 30.3	462.7 ± 28.9	472.1 ± 33.0	465.3 ± 36.2
Hb (g/dL)	Fucoidan	13.7 ± 1.6	13.8 ± 1.5	13.7 ± 1.7	13.6 ± 1.7	13.8 ± 1.7
	Placebo	14.2 ± 0.9	14.2 ± 1.0	14.2 ± 0.8	14.5 ± 1.1	14.2 ± 1.2
Ht (%)	Fucoidan	42.7 ± 4.7	42.9 ± 4.2	42.3 ± 4.2	42.5 ± 4.4	43.2 ± 4.7
	Placebo	43.1 ± 2.2	43.7 ± 3.0	43.6 ± 2.1	44.7 ± 2.8	43.8 ± 3.2
Plt (×10^4^/μL)	Fucoidan	26.2 ± 5.4	26.5 ± 5.9	26.1 ± 4.6	28.4 ± 9.4	27.4 ± 4.8
	Placebo	27.3 ± 3.5	27.4 ± 4.0	27.5 ± 4.2	27.9 ± 4.4	28.3 ± 4.3

All data represent means ± standard deviation (*n* = 20 subjects per group). WBCs: white blood cells; RBCs: red blood cell; Hb: hemoglobin; Ht: hematocrit; Plt: platelets; w: week.

**Table 4 marinedrugs-19-00340-t004:** Biochemical test results.

Parameter	Group	0 w	4 w	8 w	12 w	16 w
AST (U/L)	Fucoidan	22.4 ± 11.2	23.9 ± 17.6	21.2 ± 11.2	21.2 ± 8.8	22.7 ± 8.9
	Placebo	20.2 ± 5.9	21.3 ± 6.0	20.9 ± 7.1	21.8 ± 6.7	22.2 ± 4.8
ALT (U/L)	Fucoidan	17.9 ± 10.7	19.6 ± 17.1	16.9 ± 10.8	16.5 ± 10.0	18.7 ± 10.9
	Placebo	17.7 ± 9.6	18.3 ± 10.2	17.7 ± 7.9	20.9 ± 11.9	19.5 ± 9.3
LDH (U/L)	Fucoidan	177.5 ± 25.5	180.5 ± 28.0	176.6 ± 25.4	178.8 ± 28.6	177.3 ± 20.9
	Placebo	171.1 ± 21.6	176.0 ± 28.5	179.8 ± 29.8	173.0 ± 24.9	170.8 ± 25.1
T-bil (mg/dL)	Fucoidan	0.9 ± 0.3	0.9 ± 0.3	0.7 ± 0.2	0.8 ± 0.3	0.8 ± 0.3
	Placebo	0.9 ± 0.4	0.8 ± 0.4	0.8 ± 0.3	0.8 ± 0.3	0.7 ± 0.2
ALP (U/L)	Fucoidan	183.6 ± 51.5	180.7 ± 46.8	190.1 ± 53.4	189.9 ± 53.5	198.1 ± 55.1
	Placebo	172.4 ± 49.3	174.1 ± 47.1	175.6 ± 47.0	179.0 ± 52.5	182.6 ± 61.8
γ-GTP (U/L)	Fucoidan	41.2 ± 57.4	37.7 ± 46.6	34.4 ± 37.1	34.6 ± 31.9	32.5 ± 29.7
	Placebo	22.6 ± 15.5	23.4 ± 16.7	25.1 ± 17.3	27.1 ± 24.8	25.2 ± 18.1
CK (U/L)	Fucoidan	119.3 ± 67.2	133.9 ± 109.8	114.2 ± 47.5	122.3 ± 74.6	141.8 ± 112.0
	Placebo	122.7 ± 66.0	118.6 ± 66.5	136.3 ± 92.0	126.9 ± 88.1	114.7 ± 57.3
FBS (mg/dL)	Fucoidan	83.1 ± 8.3	82.4 ± 9.7	81.3 ± 7.4	79.1 ± 8.2	82.6 ± 12.5
	Placebo	82.8 ± 8.0	82.5 ± 7.0	82.7 ± 7.2	82.3 ± 7.8	81.0 ± 6.1
HbA1c (%)	Fucoidan	5.4 ± 0.3	5.5 ± 0.2	5.4 ± 0.3	5.4 ± 0.3	5.4 ± 0.2
	Placebo	5.4 ± 0.3	5.5 ± 0.3	5.4 ± 0.3	5.4 ± 0.2	5.4 ± 0.3
TC (mg/dL)	Fucoidan	200.7 ± 32.0	203.2 ± 29.7	201.7 ± 26.7	202.6 ± 26.6	213.9 ± 33.1
	Placebo	203.2 ± 29.8	205.3 ± 30.1	212.3 ± 33.9	215.7 ± 37.1	213.1 ± 30.8
LDL-C (mg/dL)	Fucoidan	111.1 ± 32.0	111.1 ± 31.8	109.8 ± 31.9	106.8 ± 32.4	115.5 ± 34.5
	Placebo	117.6 ± 26.7	119.2 ± 26.0	125.6 ± 29.9	126.9 ± 33.7	124.6 ± 31.6
HDL-C (mg/dL)	Fucoidan	69.8 ± 16.2	72.1 ± 17.2	68.8 ± 16.1	72.0 ± 20.1	76.4 ± 21.5
	Placebo	64.7 ± 13.1	64.7 ± 15.6	67.4 ± 13.5	66.4 ± 15.4	66.0 ± 15.7
TG (mg/dL)	Fucoidan	81.9 ± 32.0	81.3 ± 48.8	93.8 ± 48.8	95.2 ± 92.4	97.0 ± 105.7
	Placebo	98.0 ± 92.3	86.4 ± 47.7	84.8 ± 39.9	88.8 ± 39.8	96.8 ± 44.2
TP (g/dL)	Fucoidan	7.2 ± 0.4	7.2 ± 0.3	7.1 ± 0.4	7.3 ± 0.3	7.3 ± 0.3
	Placebo	7.3 ± 0.4	7.4 ± 0.4	7.3 ± 0.4	7.4 ± 0.4	7.4 ± 0.3
Alb (g/dL)	Fucoidan	4.5 ± 0.3	4.5 ± 0.2	4.4 ± 0.2	4.5 ± 0.2	4.5 ± 0.2
	Placebo	4.5 ± 0.3	4.5 ± 0.2	4.5 ± 0.2	4.5 ± 0.2	4.5 ± 0.2
UN (mg/dL)	Fucoidan	12.4 ± 4.4	12.2 ± 3.7	12.2 ± 2.9	13.1 ± 3.9	13.6 ± 3.1
	Placebo	13.1 ± 3.5	13.0 ± 3.9	12.2 ± 3.7	12.2 ± 3.7	14.3 ± 4.7
CRE (mg/dL)	Fucoidan	0.7 ± 0.1	0.7 ± 0.1	0.7 ± 0.1	0.7 ± 0.1	0.7 ± 0.1
	Placebo	0.8 ± 0.1	0.8 ± 0.1	0.7 ± 0.1	0.7 ± 0.1	0.7 ± 0.1
UA (mg/dL)	Fucoidan	5.3 ± 1.8	5.3 ± 1.6	5.1 ± 1.6	5.2 ± 1.6	5.2 ± 1.6
	Placebo	5.1 ± 1.3	5.2 ± 1.3	5.2 ± 1.3	5.3 ± 1.2	5.2 ± 1.6
Na (mEq/L)	Fucoidan	140.1 ± 1.8	140.0 ± 1.6	140.4 ± 2.1	140.8 ± 1.8	140.2 ± 1.1
	Placebo	140.4 ± 1.1	140.3 ± 1.5	141.0 ± 1.5	141.0 ± 1.8	140.7 ± 1.1
Cl (mEq/L)	Fucoidan	103.0 ± 1.6	103.1 ± 2.2	103.9 ± 1.9	103.7 ± 2.1	103.3 ± 1.9
	Placebo	103.3 ± 1.3	103.7 ± 1.6	103.9 ± 1.6	103.6 ± 2.0	104.1 ± 1.8
K (mEq/L)	Fucoidan	4.3 ± 0.3	4.2 ± 0.3	4.4 ± 0.2	4.3 ± 0.3	4.4 ± 0.2
	Placebo	4.2 ± 0.2	4.3 ± 0.4	4.3 ± 0.3	4.2 ± 0.3	4.3 ± 0.2
Ca (mg/dL)	Fucoidan	9.6 ± 0.3	9.6 ± 0.3	9.4 ± 0.3	9.6 ± 0.4	9.6 ± 0.3
	Placebo	9.7 ± 0.3	9.7 ± 0.3	9.5 ± 0.3	9.6 ± 0.3	9.6 ± 0.3
IP (mg/dL)	Fucoidan	3.6 ± 0.5	3.6 ± 0.5	3.6 ± 0.4	3.8 ± 0.5	3.6 ± 0.5
	Placebo	3.7 ± 0.4	3.7 ± 0.5	3.6 ± 0.5	3.6 ± 0.4	3.6 ± 0.6
Mg (mg/dL)	Fucoidan	2.1 ± 0.1	2.2 ± 0.2	2.1 ± 0.1	2.2 ± 0.1 ^†^	2.1 ± 0.1 ^†^
	Placebo	2.2 ± 0.1	2.2 ± 0.2	2.1 ± 0.1	2.1 ± 0.1 ^†^	2.2 ± 0.1 ^†^
Fe (μg/dL)	Fucoidan	117.1 ± 49.0	104.7 ± 35.0	112.9 ± 37.7 ^†^	108.7 ± 51.6	98.9 ± 52.6
	Placebo	99.9 ± 43.5	95.9 ± 45.2	86.4 ± 32.5 ^†^	103.2 ± 48.4	116.6 ± 37.4

All data represent means ± standard deviation (*n* = 20 subjects per group). AST: aspartate aminotransferase; ALT: alanine aminotransferase; LDH: lactate dehydrogenase; T-bili: total bilirubin; ALP: alkaline phosphatase; γ-GTP: γ-glutamyl transpeptidase; CK: creatine kinase; FBS: fasting blood sugar; HbA1c: hemoglobin A1c; TC: total cholesterol; LDL-C: low-density lipoprotein cholesterol; HDL-C: high-density lipoprotein cholesterol; TG: triglyceride; TP: total protein; Alb: albumin; UN: urea nitrogen; CRE: creatinine; UA: uric acid; Na: sodium; Cl: chlorine; K: potassium; Ca: calcium; IP: inorganic phosphorus; Mg: magnesium; Fe: iron; w: week. ^†^ Significant difference between the placebo and fucoidan groups within the same week (*p* < 0.05, *t*-test).

**Table 5 marinedrugs-19-00340-t005:** List of adverse events during the ingestion period.

Group	Adverse Event	Case	Treatment	Outcome	Causality	Study
Fucoidan	Cold	7	3	7	All unrelated	Continue all
	Pharyngitis	2	1	2	Unrelated	Continue all
	Neck strain	1	1	1	Unrelated	Continue
	Tooth cavity	1	1	1	Unrelated	Continue
Placebo	Cold	10	9	10	All unrelated	Continue all
	Pharyngitis and cough	1	0	1	Unrelated	Continue
	Abdominal pain	1	0	1	Unrelated	Continue
	Automatic nerves disorder	1	1	1	Probably unrelated	Continue
	Sprain	1	1	1	Unrelated	Continue
	Back pain	1	1	1	Unrelated	Continue
	Fatigue	1	1	1	Unrelated	Continue
	Gum disease	1	1	1	Unrelated	Continue

Numbers represent the number of subjects.

## Data Availability

The data are not publicly available due to privacy.

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
