# Peer review of "Effects of Ingesting Fucoidan Derived from Cladosiphon okamuranus Tokida on Human NK Cells: A Randomized, Double-Blind, Parallel-Group, Placebo-Controlled Pilot Study"

_marinedrugs, 2021, doi:10.3390/md19060340_

Round 1
Reviewer 1 Report
This manuscript described the administration of 3g/d of fucoidin to 20 volunteers over 16w and compared to 20 health controls. The authors conclude that the dietary supplement increased NK cell activity in males at the 4, 8 and 12w times points (but not 16w after stopping treatment at 12w). It is understandable that the small number of participants may lead to results with a high degree of variability, however, in this case the conclusions are not supported by the results and the methods are inappropriate and/or inadequately described.
Major concerns:
- The description of NK cell isolation and activity analysis is lacking key details, including how were NK cells isolated and what does "activity" mean in this study? Individual measurements, not averages, should be shown on the charts. Furthermore, it is not stated what the error bars on the charts indicate. Based on the width of these bars, it appears unlikely the values at 4w, 8w and 12w are significantly different from 0w.
-
The NK subset analysis suggests the majority of these aren't NK cells, because 75%+ are CD57-/CD16-, thus the decrease in CD57+/CD16+ is not considered appropriate. Furthermore, because the isolation of NK cells is not described, the data in Table 6 cannot be properly evaluated. Are these PBMCs? In which case NK the NK cell subset is surprisingly high. Also, the % of cells CD16+ is very low for primary human NK cells. Lastly, the use of CD57 is surprising, CD56 would be a better marker for NK cells.
Minor Concerns:
- The administration rate of 3g/d is buried in the discussion and not mentioned in the Results or the Procedures sections. This is a serious oversight.
- It is unclear why Table 1 contains information on a study participant that did not complete the study. It is also not mentioned when the sole participant withdrew.
Author Response
Dear Reviewer,
We thank the referees for carefully reading our manuscript and providing valuable comments and suggestions. In response to the reviewers’ comments, we have revised the manuscript.
We hope that our manuscript is now acceptable for publication in Marine Drugs.

Reviewer 2 Report
Tomori and colleagues here study the role of fucoidan, a sulfated polysaccharides contained in brown algae, specifically derived from Okinawa mozuku (Cladosiphon okamuranus) on human immune cells, particularly NK cells but also neutrophils. Fucoidan interestingly increases NK activation, especially in male individuals, while regulation of neutrophil activity seems to be more complex depending on the effector function.
Together this study provides some originality to the audience especially since it points to an immune activating drug that could be ingested to boost innate immunity while there appear to be little to no related side effects.
However, some clarifications will be needed to substantiate the strategy and major findings in this study.
Major comments
Please comment on choosing CD57 and CD16 as markers for NK cells, since this selection does not cover the whole NK cell fraction (e.g. missing immature NK cells).
Table 2, 3, 5, 6 contain major findings that cannot be fully appreciated with just giving the numbers. Please generate graphs connecting the respective parameters during the time course analyzed and comment on the longitudinal effects.
How was the dosage / regimen of fucoidan ingestion (3 g) for 12 weeks chosen. Are there comparative data available to other regimen?
Please include a discussion on the different effects of NK cells from male and female donors.
A direct proof of altered production of IFNg by NK cells in the presence of fucoidan should be provided at least in vitro.
Minor comments
line 43
the authors say they reported in reference 17, although no common authors are detectable in that study. Please check.
Author Response
Dear Reviewer, We thank all the referees for carefully reading our manuscript and providing valuable comments and suggestions. In response to the reviewers’ comments, we have revised the manuscript. We hope that our manuscript is now acceptable for publication in Marine Drugs.

Reviewer 3 Report
The manuscript aims at evaluating the effects of ingesting fucoidan derived from Okinawa mozuku (Cladosiphon okamuranus) on natural killer (NK) cell activity. This could be envisaged as a potential dietary intervention to activate immune cell response, here specifically NK cells, for example in cancers, using a well tolerable approach.
I have some comments and suggestions about the result presentation and interpretation:
1) Sample size. The authors used only 20 subjects as a placebo and 20 subjects (treatments). These sample sizes are too small for such a study. Therefore, I suggest considering this point to be discussed. Already in the title, I would define the work presented as a "pilot" study.
2) Markers of NK cells activation. The authors used CD57 and CD16 as a readout of NK cell activation. Representative dot plots should be shown. CD57 and CD16 characterize mature NK, non-necessarily activated, therefore I would suggest to revised these concepts accordingly.
3) NK cell activity was detected by 51Cr-labeled target cell assay. Please, specified which target cells? K562? Did the authors perform the CD107a (degranulation) assay? These experiments, together with IFNg and Perforin production, by NK cells to the two different experimental groups, should be performed, to support NK cell activation.
4) Authors should better contextualize the analysis on neutrophils within the study (this part is totally missed in the discussion). There are no details on how neutrophils have been identified by FACS? Which CDs have been used? Gating strategy? Since the authors used whole blood and not isolated/sorted neutrophils, representative dot plots are necessary to show the signals of 2’,7’-dichlorofluorescein-diacetate specifically in neutrohils.
Based on my comments, I would consider the manuscript to be accepted for publication, pending major revisions.
Author Response
Dear Reviewer, We thank the referees for carefully reading our manuscript and providing valuable comments and suggestions. In response to the reviewers’ comments, we have revised the manuscript.

Round 2
Reviewer 1 Report
The authors addressed the major concerns, and though the effect appears minor, suitable data are provided to allow others to thoroughly review the results.
Author Response
Dear reviewer 1,
Thank you very much for providing important comments.
We are thankful for time and energy you expended.
We feel that the comments have allowed us to improve the manuscript.
Best Regards,
Makoto Tomori
Reviewer 2 Report
Tomori and colleagues have addressed all comments by the reviewer.
Two minor issues still need to be clarified.
please check wording lines 183-185 "This result suggests that fucoidan supplementation induces NK cell activity, reducing immunity caused by estrogen deficiency."
please specify lines 253f: "The 51Cr released from the target cells by NK cell cytotoxicity was determined using a radiation measurement instrument"
Author Response
Dear reviewer 2,
Thank you very much for providing important comments.
We are thankful for time and energy you expended.
Our responses to the reviewer’s comments are as attached file.
